# Sparse Bayesian Relevance Vector Machine Identification Modeling and Its Application to Ship Maneuvering Motion Prediction

**Yao Meng** [1,2], **Xianku Zhang** [1,2,*] , **Guoqing Zhang** [1] , **Xiufeng Zhang** [1,2] **and Yating Duan** [1,2]

1   Navigation College, Dalian Maritime University, Dalian 116026, China
2   Key Laboratory of Marine Simulation and Control, Dalian Maritime University, Dalian 116026, China
*   Correspondence: zhangxk@dlmu.edu.cn

**Abstract:** In order to establish a sparse and accurate ship motion prediction model, a novel Bayesian probability prediction model based on relevance vector machine (RVM) was proposed for nonparametric modeling. The sparsity, effectiveness, and generalization of RVM were verified from two aspects: (1) the processed Sinc function dataset, and (2) the tank test dataset of the KRISO container ship (KCS) model. The KCS was taken as the main research plant, and the motion prediction models of KCS were obtained. The $\varepsilon$-support vector regression and $v$-support vector regression were taken as the compared algorithms. The sparsity, effectiveness, and generalization of the three algorithms were analyzed. According to the trained prediction models of the three algorithms, the number of relevance vectors was compared with the number of support vectors. From the prediction results of the Sinc function and tank test datasets, the highest percentage of relevance vectors in the trained sample was below 17%. The final prediction results indicated that the proposed nonparametric models had good prediction performance. They could ensure good sparsity while ensuring high prediction accuracy. Compared with the SVR, the prediction accuracy can be improved by more than 14.04%, and the time consumption was also relatively lower. A training model with good sparsity can reduce prediction time. This is essential for the online prediction of ship motion.

**Keywords:** nonparametric identification modeling; ship motion prediction; relevance vector machine; sparsity analysis; support vector regression

## 1. Introduction

### 1.1. Background and Motivation

The research on maritime autonomous surface ships (MASS) is widely discussed [1]. More intelligent ships are being used for motion control and sea transportation. In recent years, with the construction of MASS, the accuracy of the model directly affects the performance of the motion control. The sparse modeling method can avoid overfitting to a certain extent and reduce the prediction time [2], which can lay the foundation of online prediction. Ship motion prediction is of great significance for the design of course-keeping, path-following, and adaptive controllers, and it can also improve the control effect [3,4]. Therefore, the development of an efficient and sparse modeling method is of great importance for identification modeling and overcoming the effect of ship type, etc.

### 1.2. Related Work

System identification has been widely used in many research fields, which solves the modeling problem. It includes parameter and nonparametric identification. In the field of parameter identification of ship motion models, some studies have been conducted using classical and intelligent algorithms. Classical algorithms, such as least square (LS) [5], extended Kalman filter (EKF) [6,7], gradient descent (GD) [8], and recursive prediction error (RPE) [9], were used for the parameter estimation of MMG, coupled with heave-pitching

motion models. In the parameter estimation results, these algorithms ensured certain estimation accuracy. As for intelligent algorithms, neural network (NN) [10] and support vector machine (SVM) [11,12] were frequently applied to the parameter estimation of ship motion models. For machine learning algorithms, a certain amount of data is required as support to obtain accurate parameter estimation results. A sensitivity analysis [13,14], as a way to simplify models, reduced the complexity of parameter identification.

As for nonparametric identification modeling, intelligent identification algorithms are widely used. With the rapid development of artificial intelligence, NNs with different structures have been proposed. Recurrent NN [15–17], self-designed fully connected NN [18], and long short-term memory NN [19] were applied to the maneuvering motion prediction of KCS, KVLCC2, and DTMB 5415, respectively. Moreover, compared to the traditional NN, the emerging NN can obtain high prediction accuracy. Based on statistical learning theory, SVM uses the kernel method to avoid dimension disaster. It includes $\varepsilon$-SVR and $v$-SVR when it is used for regression. When $\varepsilon$-SVR [20] and $v$-SVR [21] were applied to the parameter identification, precise ship motion prediction results were obtained. However, the setting of hyperparameters greatly affects the regression accuracy of SVR. Zhang et al. [22] combined the grey wolf optimizer with $\varepsilon$-SVR. Based on the full-scale trial data, the hyperparameters of $\varepsilon$-SVR were tuned and more accurate prediction models are obtained. The supervised learning algorithm, fast noisy input Gaussian process (GP) [23], was used to conduct online motion prediction for container ships and unmanned surface ships. Local weighted learning (LWL) was widely used in robot control fields and is an effective identification algorithm for ship motion prediction [24,25]. As a deep learning algorithm, delayed deep cycle reservoir with regular jumps (DDCRJ) [26] was used to predict ship heave motion. Table 1 shows a summary of relevant work.

**Table 1.** Summary of related work.

| System Identification | Algorithm Types | Algorithms | Advantages | | Disadvantages | |
|---|---|---|---|---|---|---|
| Parameter identification | Classical algorithms | LS [5] EKF [6,7] GD [8] RPE [9] | (1) (2) (3) | Each step takes less time; Simple structure; Mathematical reasoning, etc. | (1) (2) (3) | The influence of the initial value; Parameters drift; Convergence speed. |
| | Intelligent algorithms | NN [10] SVM [11,12] | (1) (2) (3) | Generalization of parameters; Learning ability; Convergence speed, etc. | (1) (2) (3) | Training time; The influence of the data sample; Complex structure. |
| Nonparametric identification | Intelligent algorithms | NN [15–19] SVM [20–22] GP [23] LWL [24,25] DDCRJ [26] | (1) (2) (3) | Generalization of prediction model; Prediction accuracy; Wide range of applications. | (1) (2) (3) | The influence of the data sample; Long training time; Hyperparameter tuning. |

Most algorithms used in Table 1 were applied to three degrees of freedom (3 DOF) ship motion, response, and coupled heave-pitching motion models. For the current ship motion controller design, most scholars used the ship response and 3 DOF motion models. Therefore, there are many studies on the 3 DOF ship motion model. With the development of MASS, the prediction of their plane motion is also necessary. SVM has featured in a lot of research in the fields of both parameter identification and nonparametric identification. It has led to good research results. However, although SVM is a relatively sparse model, the computational complexity will increase significantly as the training sample size becomes larger. RVM has many of the features of SVM and avoids the main limitations of SVM. In addition, compared to the state-of-the-art ship motion prediction models, there are relatively few studies on algorithmic sparsity.

Sparsity encompasses feature sparsity and model sparsity. Feature sparsity refers to the sparsity of feature vectors. Model sparsity refers to the sparsity of model weights. This study aims to explore the model sparsity of the three algorithms while ensuring prediction accuracy. The obtained regression model has a certain sparsity if the weights of the regression function have more zeros. Sparsity can significantly increase model capacity and performance without a proportional increase in computation. RVM combines the kernel method with Bayes' theorem to obtain a relatively sparse prediction model. Taking RVM as an example, the number of relevance vectors (RVs) in the prediction model is small (i.e., the weights of the nonparametric model obtained by RVM have more zeros). RVM only has one hyperparameter to be tuned. Therefore, it does not require an optimization algorithm to tune the hyperparameter. It can quickly obtain a suitable hyperparameter by trial and error, and also guarantee good prediction accuracy. The related work on RVM is discussed as follows.

Sparse models are more and more widely used in many research fields, which can avoid overfitting in the modeling process. RVM has been applied in many fields [27–29]. Based on RVM, Tian et al. [30] proposed a diagnosis method for high-pressure hydrogen leakage. In some specific cases, the proposed model took less time. Moreover, the proposed diagnostic model had an accuracy rate of more than 95%. Chang et al. [31] used an empirical model to predict the tunnel convergence. The prediction residuals were modified by RVM. The probability model for tunnel convergence monitoring was obtained. Compared to the prediction results of BPNN and GP, the proposed algorithm had good prediction performance.

### 1.3. Contributions and Structure

Based on the descriptions in Sections 1.1 and 1.2, studies of the sparsity and prediction performance of RVM and SVR are conducted. This research proposes a sparse ship motion prediction model based on RVM. The sparse model removes many redundant variables, simplifying the prediction models while retaining the most significant information in the dataset. Two different datasets, the Sinc function dataset and tank test dataset of KCS, are taken as illustrative examples to analyze the sparsity, effectiveness, and generalization of RVM, $\varepsilon$-SVR, and $v$-SVR. The novelty of the research lies in the following three points:

(1) Unlike most existing machine learning algorithms, RVM is a sparse model proposed under the fast sequential sparse Bayesian framework. This can be used to obtain a sparse ship motion prediction model.

(2) Based on the Bayesian framework, a regularization parameter is not used in the derivation of RVM. This reduces the complexity of model learning. Additionally, the obtained ship motion prediction models can avoid overfitting.

(3) The kernel function of RVM does not need to satisfy the Mercer condition. RVM can flexibly choose the kernel function to obtain an accurate ship motion prediction model. For the setting of the hyperparameter, RVM only needs to set the kernel function parameter.

The structure of the research is organized as follows: Section 2 describes the framework of a 3 DOF ship maneuvering motion model and gives the framework of a nonparametric identification model based on RVM. Section 3 introduces Bayesian inference, sparse Bayesian learning, and the derivation and principle of RVM. In Section 4, the research introduces two illustrative examples to demonstrate the sparsity, effectiveness, and generalization of the nonparametric identification model. Section 5 summarizes the conclusions and future research prospects.

## 2. Ship Maneuvering Motion Model

In order to describe the ship motion, earth-fixed coordinates and body-fixed coordinates are adopted. Figure 1 shows these two coordinate systems.

The main objective is to obtain a 3 DOF nonparametric identification model of KCS. Therefore, the 3 DOF motion model of KCS is established [32]. The Abkowitz model is

widely used in identification modeling research. Although its derivation process is relatively complex, it can accurately describe the dynamic maneuvering characteristics of the ship [33]. Moreover, identification modeling simplifies the calculation of ship hydrodynamic derivatives, so as to establish an accurate mathematical model of ship motion based on data samples. Therefore, the Abkowitz model is used in this study. Equation (1) lists the expression of the 3 DOF motion model for KCS.

$$
\begin{cases}
m\left(\dot{u} - vr - x_G r^2\right) = X \\
m\left(\dot{v} + ur + x_G \dot{r}\right) = Y \\
I_z \dot{r} + m x_G \left(\dot{v} + ur\right) = N \\
\dot{\psi} = r \\
\dot{x} = u\cos(\psi) - v\sin(\psi) \\
\dot{y} = u\sin(\psi) + v\cos(\psi)
\end{cases}
\tag{1}
$$

where $m$ is the mass of ship; $x_G$ is the gravity center of ship; $X$, $Y$, and $N$ are force and moment; $I_z$ is the added moment of inertia; $u$, $v$, $r$, and $\psi$ are surge velocity, sway velocity, yaw rate, and yaw angle, respectively; $x$ and $y$ are displacement in $X_0$ and $Y_0$ direction.

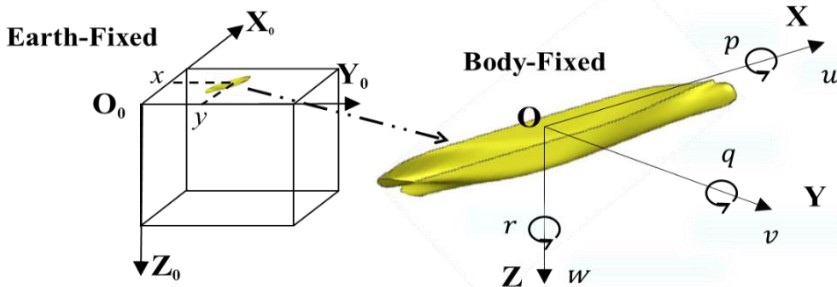

**Figure 1.** Coordinate systems.

The Equation (1) is expressed by the state space equation to obtain Equation (2).

$$
\begin{bmatrix}
m - X_{\dot{u}} & 0 & 0 \\
0 & m - Y_{\dot{v}} & m x_G - Y_{\dot{r}} \\
0 & m x_G - N_{\dot{v}} & I_{zz} - N_{\dot{r}}
\end{bmatrix}
\begin{bmatrix}
\dot{u}(t) \\
\dot{v}(t) \\
\dot{r}(t)
\end{bmatrix}
=
\begin{bmatrix}
f_X(t) + \tau_X(t) \\
f_Y(t) + \tau_Y(t) \\
f_N(t) + \tau_N(t)
\end{bmatrix}
\tag{2}
$$

where $X_{\dot{u}}$, $Y_{\dot{v}}$, $Y_{\dot{r}}$, $N_{\dot{v}}$, and $N_{\dot{r}}$ are ship hydrodynamic derivatives; $f_1$, $f_2$, and $f_3$ are the nonlinear functions related to force and moment; $\tau_X$, $\tau_Y$, and $\tau_N$ are forces and moment coming from environmental disturbances.

Nonparametric identification modeling considers the input, output, and dynamic process of the system, without considering the internal structure of the model. According to the motion model of KCS established by Equation (2), the mapping relationship between the ship motion is shown in Equation (3).

$$
\begin{cases}
\dot{u}(t_{n+1}) = \mathrm{RVM}_1(u(t_n), v(t_n), r(t_n), \delta(t_{n+1})) \\
\dot{v}(t_{n+1}) = \mathrm{RVM}_2(u(t_n), v(t_n), r(t_n), \delta(t_{n+1})) \\
\dot{r}(t_{n+1}) = \mathrm{RVM}_3(u(t_n), v(t_n), r(t_n), \delta(t_{n+1}))
\end{cases}
\tag{3}
$$

where $\delta$ is the rudder angle and $\mathrm{RVM}_1$, $\mathrm{RVM}_2$, and $\mathrm{RVM}_3$ are mapping function.

## 3. Methodology

### 3.1. Bayesian Inference and Sparse Bayesian Learning

Suppose the system observation data set is $D$. $\theta$ is the system parameter. Then, the likelihood function of the system is $p(D|\theta)$, and the prior distribution of parameters is $p(\theta)$.

According to Bayes' theorem, the posterior distribution $p(D|\theta)$ of the system is Equation (4).

$$p(\theta|D) = \frac{p(\theta)p(D|\theta)}{p(D)} \tag{4}$$

where $p(D)$ is the marginal likelihood function.

Bayesian inference uses probability operations to fit observed data. $x_p$ is the new data to be predicted. According to Bayesian inference, the probability distribution of the predicted points is calculated as Equation (5).

$$p(x_p|D) = \int p(x_p, \theta|D)d\theta = \int p(x_p|\theta, D)p(\theta|D)d\theta \tag{5}$$

Sparse Bayesian learning combines the prior information of unknown parameters with sample information to obtain the assumption of the prior probability distribution [34]. Then, the posterior information is obtained according to Bayes' theorem. Finally, the obtained posterior information is used to update the prior distribution and sample information. After the above multiple iterations, more and more accurate sample information is obtained. In general, the model of sparse learning can be expressed as Equation (6).

$$y_s(x) = \Phi_s(x_s)w_s \tag{6}$$

where $y_s(x)$ is the regression value of sparse model, $\Phi_s(x_s)$ is the kernel functions matrix, $w_s$ is the weight, and $x_s$ is the training sample.

The basic idea of sparsity is to use the weighted idea to represent a test sample after the training sample is mapped linearly or nonlinearly. Since the test samples are only related to the training samples with similar information, most of the weighted values are 0 to achieve the purpose of sparsity.

### 3.2. Relevance Vector Machine

In the 1990s, the rise of SVM led to research into the kernel method. The kernel method is regarded as the third revolution of pattern analysis method after classical statistical linear analysis, NN, and decision tree nonlinear analysis [35]. Therefore, the sparsity of SVM after the introduction of the kernel method has captured the interest of scholars. With the in-depth study of SVM, the defects of SVM have been gradually revealed.

RVM is a supervised learning algorithm proposed by Tipping [36]. Based on Bayesian inference and the sparse Bayesian learning idea, it has many features of SVM and avoids its main limitations. Compared with land vehicles, ships have the characteristics of delayed response to maneuvering. Therefore, there are complex mapping relationships between data. The sparse prediction model can reduce this effect and improve the generalization performance. Given a set of the KCS motion dataset as $\{x_n, t_n \mid n = 0, 1, 2, \ldots, N\}$, $N$ is the number of data samples. According to probability theory, $t_n$ can be expressed as Equation (7). The regression function $f(\mathbf{x})$ is shown in Equation (8).

$$t_n = y(x_n, \boldsymbol{w}) + \varepsilon_n, \varepsilon_n \sim \mathcal{N}\left(0, \sigma^2\right) \tag{7}$$

$$f(\mathbf{x}) = \sum_{n=1}^{N} \omega_n K(\mathbf{x}, x_n) + b \tag{8}$$

where $\boldsymbol{w}$ is a vector of the weight parameters, $\varepsilon_n$ is the random noise, $\mathcal{N}$ is the normal distribution, and $b$ is the bias.

Supposing that $t_n$ is a mutually independent distribution, the likelihood estimation probability of the training dataset is shown in Equation (9).

$$\mathcal{P}\left(\boldsymbol{t}|\boldsymbol{w}, \sigma^2\right) = \left(2\pi\sigma^2\right)^{-\frac{N}{2}} \exp\left(-\frac{\|\boldsymbol{t} - \boldsymbol{\Phi}\boldsymbol{w}\|^2}{2\sigma^2}\right) \tag{9}$$

where $t = (t_1, t_2 \ldots, t_N)^{\text{T}}$, $\Phi$ is the $N \times (N+1)$ kernel functions matrix, i.e., $\Phi = [\varphi(x_1), \varphi(x_2), \ldots, \varphi(x_n)]^{\text{T}}$, $\varphi(x_i) = [1, K(x_i, x_1), K(x_i, x_2) \ldots K(x_i, x_n)]$.

If the maximum likelihood estimation of weights is directly conducted, overfitting may occur. To avoid overfitting, it is necessary to constrain $w$ in the RVM regression model. Suppose that $w$ is mutually independent and it follows $\mathcal{N}(0, \sigma^2)$. The expression of prior probability is shown in Equation (10).

$$P(w|\alpha) = \prod_{i=0}^{N} \mathcal{N}\left(\omega_i \,\middle|\, 0, \alpha_i^{-1}\right) \tag{10}$$

where $\alpha$ is a vector of $N + 1$ hyperparameters.

According to the prior probability and maximum likelihood estimation, the posterior probability of $w$ can be obtained. The expression of posterior probability is shown in Equation (11). Maximize $P(t|\alpha, \sigma^2)$ to obtain Equation (12).

$$P\left(w\,\middle|\,t, \alpha, \sigma^2\right) = \frac{P(t|w, \sigma^2) P(w|\alpha)}{P(t|\alpha, \sigma^2)} = -(2\pi)^{-\frac{N+1}{2}} |\Sigma|^{-\frac{1}{2}} \exp\left(-\frac{1}{2}(w-\mu)^{\text{T}} \Sigma^{-\frac{1}{2}}(w-\mu)\right) \tag{11}$$

$$P\left(t\,\middle|\,\alpha, \sigma^2\right) = (2\pi)^{-\frac{N}{2}} |\Omega|^{-\frac{1}{2}} \exp\left(-\frac{1}{2} t^{\text{T}} \Omega t\right) \tag{12}$$

The expression of posterior mean and covariance is shown in Equation (13).

$$\begin{cases} \mu = \sigma^{-2} \Sigma \Phi^{\text{T}} t \\ \Sigma = \left(\sigma^2 \Phi^{\text{T}} \Phi + \mathbf{A}\right)^{-1} \end{cases} \tag{13}$$

where $\mathbf{A} = \text{diag}(\alpha_0, \alpha_1 \ldots \alpha_N)$, representing the diagonal matrix about $\alpha$. $\Omega = \sigma^2 \mathbf{I} + \Phi \mathbf{A}^{-1} \Phi^{\text{T}}$.

The expressions for hyperparameter updating are shown in Equations (14) and (15).

$$\alpha_i^{New} = \frac{\gamma_i}{\mu_i^2} \tag{14}$$

$$\sigma_i^{New} = \frac{\|t - \Phi\mu\|^2}{N - \sum_{i=0}^{N} \gamma_i} \tag{15}$$

where $\gamma_i = 1 - \alpha_i \Sigma_{ii}$. $\Sigma_{ii}$ is $i$th diagonal element of the posterior covariance matrix $\Sigma$.

Supposing that the test sample is $x^*$, Equation (16) lists the predicted value $y^*$. The prediction variance $\sigma^*$ is shown in Equation (17). Equation (18) lists the probability distribution of prediction value $t^*$.

$$y^* = \mu^{\text{T}} \Phi\left(x^*\right) \tag{16}$$

$$\sigma_*^2 = \sigma_{\text{MP}}^2 + \Phi\left(x^*\right)^{\text{T}} \Sigma \Phi\left(x^*\right) \tag{17}$$

$$P\left(t^*\,\middle|\,t, \alpha_{\text{MP}}, \sigma_{\text{MP}}^2\right) = \int P\left(t^*\,\middle|\,w, \sigma_{\text{MP}}^2\right) P\left(w\,\middle|\,t, \alpha_{\text{MP}}, \sigma_{\text{MP}}^2\right) dw = \mathcal{N}\left(t^*\,\middle|\,y^*, \sigma_*^2\right) \tag{18}$$

where $\alpha_{\text{MP}}$ and $\sigma_{\text{MP}}^2$ are the optimal parameter estimates of $\alpha$ and $\sigma^2$.

**Remark 1.** *RVM combines kernel methods and Bayes' theorem to obtain a sparse probabilistic prediction model. During the derivation of the algorithm, it does not add regularization coefficients, but uses the form of a probabilistic description. In addition, as a machine learning algorithm with strong sparsity and generalization, it has been proven to have certain universality [29–31].*

## 4. Illustrative Examples and Discussion

Both $\varepsilon$-SVR and $v$-SVR show good prediction performance in the field of ship motion prediction [20,21]. However, SVR needs to determine hyperparameters, such as the penalty factor (*C*). RVM is developed based on SVR. It uses Bayes' theorem to obtain the probability prediction model without determining *C*. Because RVM has only one hyperparameter to be tuned, it does not require an optimization algorithm to tune the hyperparameter, unlike SVR. The tuning of hyperparameters depends on the published research results and the swarm intelligence optimization algorithm. This study chooses the former. RVM's feature is sparsity, which can improve prediction performance and efficiency. In the aspect of regression, the prediction model obtained by RVM is generally sparser than that obtained by SVR. Therefore, the speed can be improved when predicting test datasets. The structure of this research for RVM identification modeling is shown in Figure 2.

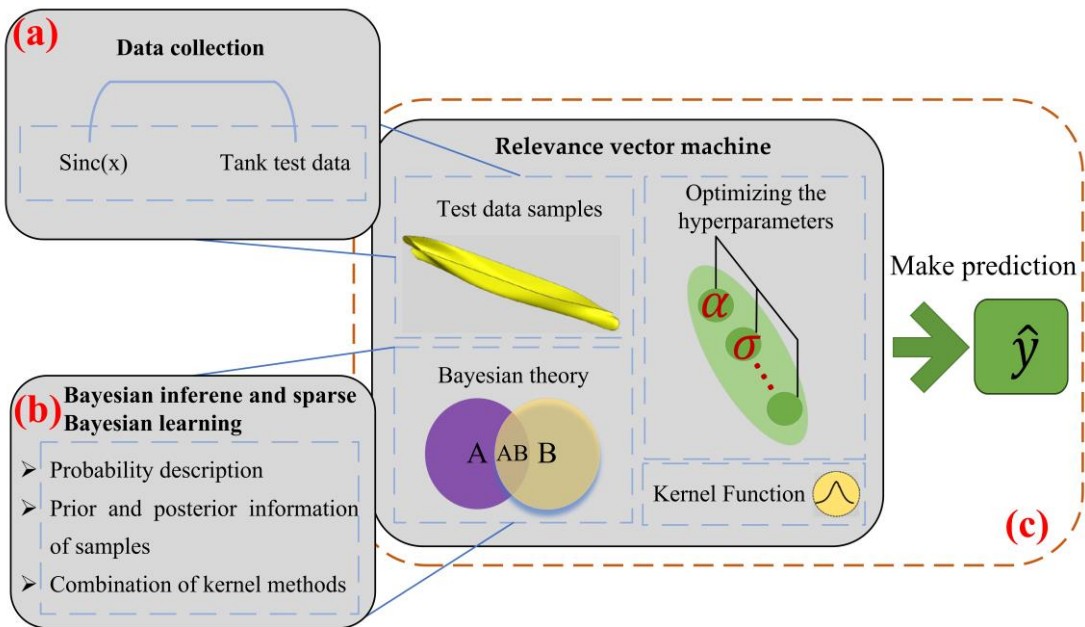

**Figure 2.** Structure of RVM identification modeling. (**a**) is data collection for two illustrative examples. (**b**) is the derivation of Bayesian inference and sparse Bayesian learning. (**c**) is the prediction process of RVM.

In order to demonstrate the effectiveness, generalization, and sparsity of the proposed algorithm, two illustrative examples are used in this research. One is the Sinc function that is often used in the fields of digital signal processing and Fourier transform theory and mathematics. The other is the KCS motion datasets published by the SIMMAN workshop in 2020 [37]. The SIMMAN workshop is used to assess simulation methods for ship maneuvering. The technical route of this research is shown in Figure 3.

### 4.1. Prediction Results of the Sinc Function Dataset

To validate the sparsity and prediction accuracy of the RVM, the Sinc function dataset with added interference is used as the verification sample. The interference is white noise with a mean of 0 and a variance of 0.02. Equation (19) lists the expression of the Sinc function. When $x = 0$, define Sinc(0) = 1. Take the independent variable $x$ as the input data and the dependent variable $y$ as the output data. The independent variable and dependent variable are scalars. The value range of $x$ is $[-20, 20]$. The sampling interval is 0.2. The number of data samples is 201. A processed data sample is shown in Figure 4.

$$y = \text{Sinc}(x) = \frac{\sin(x)}{x} \qquad (19)$$



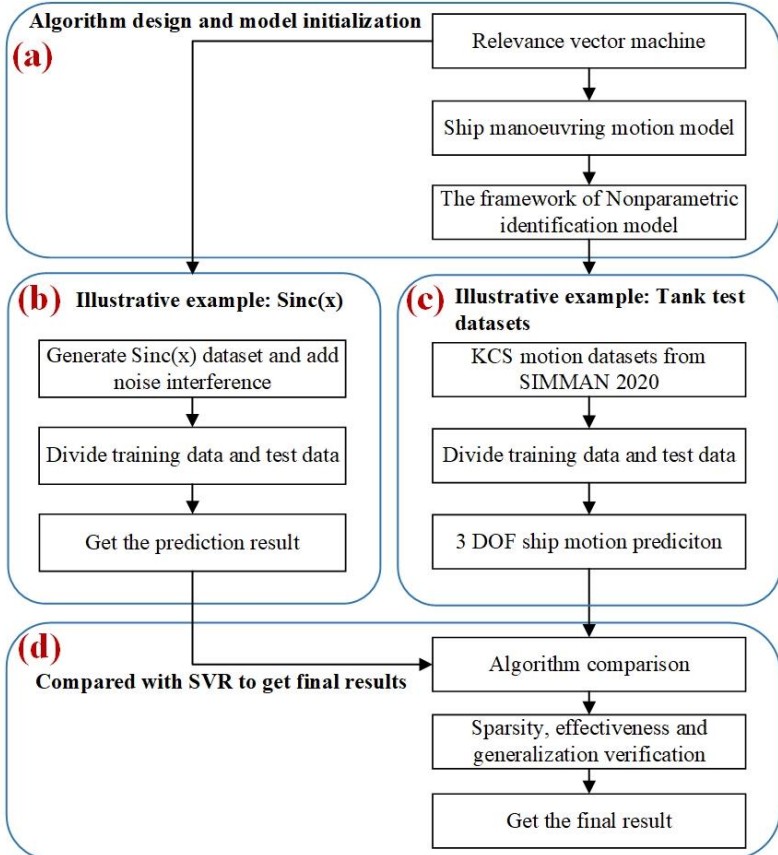

**Figure 3.** Technical route.

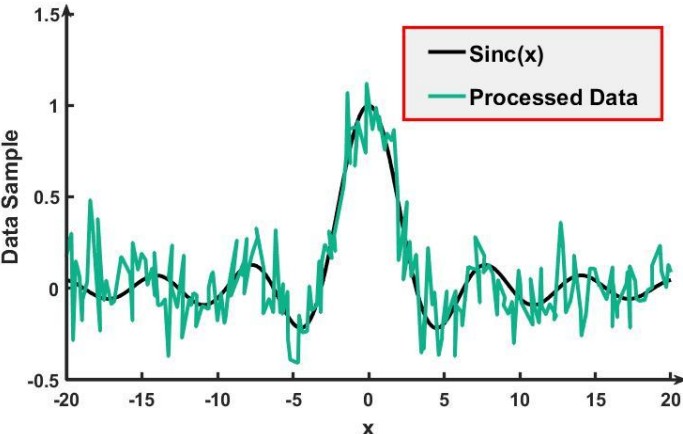

**Figure 4.** Processed data sample.

In order to reasonably divide the training and test datasets, this research takes the odd numbered data samples as the training dataset and the even numbered samples as the test dataset. The number of training and test datasets is 101 and 100, respectively. In general, the training and test sets of the Sinc function dataset are divided in a 70%/30% ratio. Because three algorithms all have small sample learning ability, this study divides the Sinc function dataset into training and test datasets in the 50%/50% ratio. The training dataset is different from the test dataset. Figure 4 shows that the processed data sample has certain noise interference. For $\varepsilon$-SVR, the number of support vectors (SVs) depends on the setting of the insensitive factor ($\varepsilon$). In general, the smaller the $\varepsilon$, the higher the prediction performance. However, a decrease in $\varepsilon$ will cause an increase in the number of SVs and low calculation

efficiency. For $v$-SVR, the setting of the parameter $v$ can adaptively adjust the number of SVs without adjusting the $\varepsilon$ [21]. To compare the experiment results, the three algorithms are set to different hyperparameter values [20,38]. In addition, compared with SVR, RVM only needs to set the kernel function parameter ($\gamma$). In terms of tuning hyperparameters, RVM also shows certain advantages. For the three algorithms, this research mainly designed three different experiments (as shown in Table 2). The experiment results of three algorithms are shown in Figure 5.

**Table 2.** Settings for three different experiments.

| Algorithms | Experiment (1) | Experiment (2) | Experiment (3) |
|---|---|---|---|
| RVM | $\gamma = 0.1$ | $\gamma = 0.5$ | $\gamma = 1.0$ |
| $\varepsilon$-SVR | $C = 10^4$, $\gamma = 0.1$, $\varepsilon = 0.01$ | $C = 10^5$, $\gamma = 0.5$, $\varepsilon = 0.1$ | $C = 10^5$, $\gamma = 0.8$, $\varepsilon = 10^{-5}$ |
| $v$-SVR | $C = 10^4$, $\gamma = 0.1$, $v = 0.5$ | $C = 10^5$, $\gamma = 0.5$, $v = 0.3$ | $C = 10^5$, $\gamma = 0.8$, $v = 0.2$ |

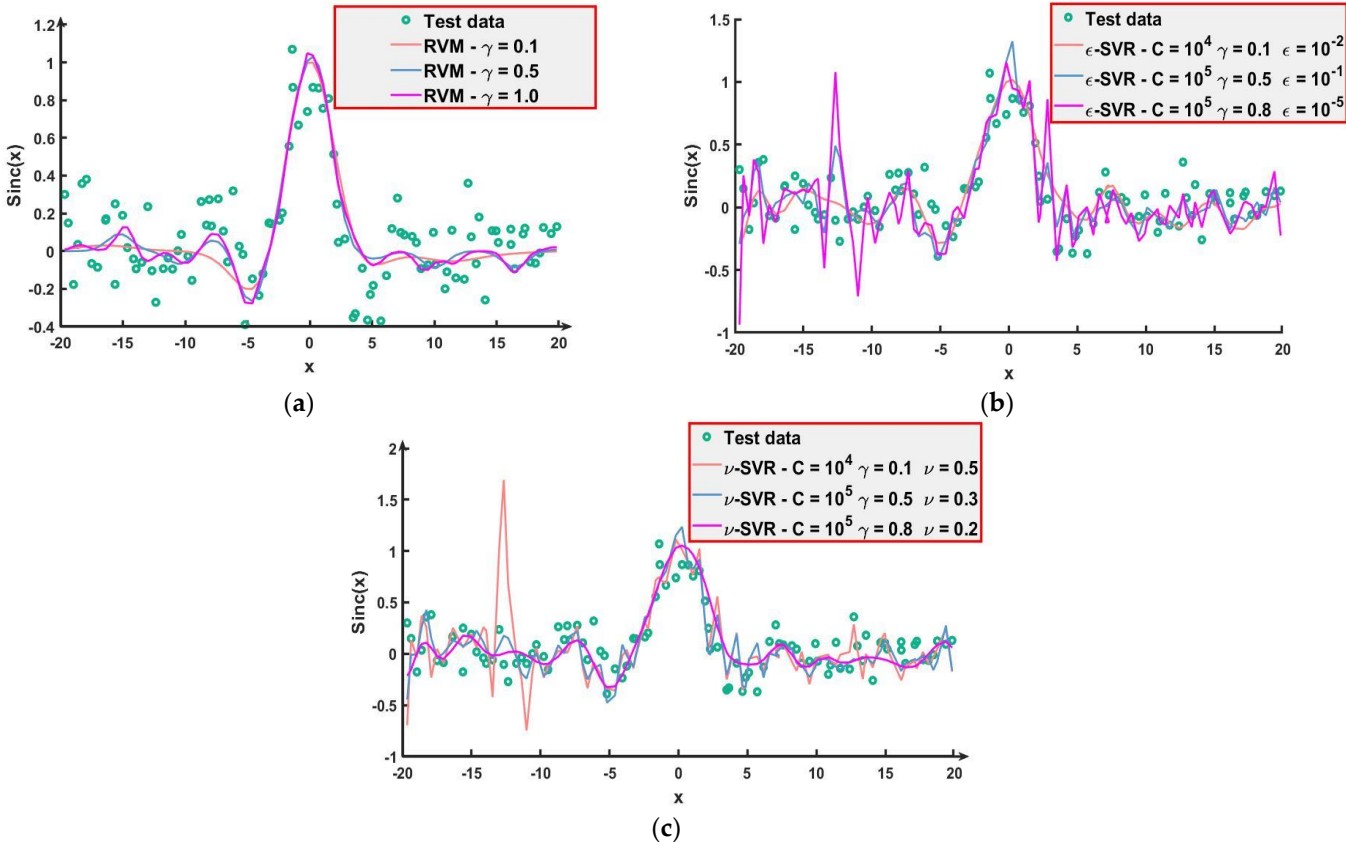

**Figure 5.** Prediction results of three algorithms for Sinc function dataset: (**a**) is the prediction result of RVM; (**b**) is the prediction result of $\varepsilon$-SVR; (**c**) is the prediction result of $v$-SVR.

Figure 5a shows that the prediction results of three experiments are smooth and fit the Sinc(x) test dataset for RVM well. In contrast, since three hyperparameters need to be set, the prediction accuracy of SVR is restricted. As can be seen from Figure 5b,c, when the hyperparameters are set differently, the prediction results of SVR differ greatly. Table 3 shows the number of RVs and SVs in three experiments. To test the sparsity of the trained prediction model, the number of RVs and SVs in experiment (2) is compared. The comparison result is shown in Figure 6.

Figure 6 shows that the number of RVs is less than the number of SVs in experiment (2). In the trained model, it can be seen from Table 3 that the number of RVs is far less than the number of SVs for the three experiments. The average percentage of RVs is 11.56%.

The average percentage of SVs is at least 6.8 times that of RVs. In addition, the highest percentage of RVs in the training sample is 16.83%. Compared with $\varepsilon$-SVR and $v$-SVR, the results verify the sparsity of RVM, i.e., RVM can attain sparser prediction models. In terms of time consumption, RVM spends a total of 0.286 s on training and predicting. However, SVR takes relatively more time, especially $v$-SVR. The mean square error (MSE) is used as the evaluation criteria to verify the prediction performance. Table 4 lists the MSEs of prediction results for the three experiments. As can be seen from Table 4, the three algorithms all have good prediction effect in the Sinc function dataset, but the prediction accuracy obtained by RVM is better compared with SVR.

**Table 3.** The number of RVs and SVs for Sinc function dataset.

| Algorithms | Experiment (1) | Experiment (2) | Experiment (3) | Average Percentage [c] |
|---|---|---|---|---|
| RVM [a] | 6 | 12 | 17 | 11.56% |
| $\varepsilon$-SVR [b] | 99 | 72 | 101 | 89.77% |
| $v$-SVR [b] | 69 | 80 | 91 | 79.21% |

[a]: The proposed algorithm; [b]: Algorithm for comparison; [c]: Average percentage of RVs and SVs in training sample.

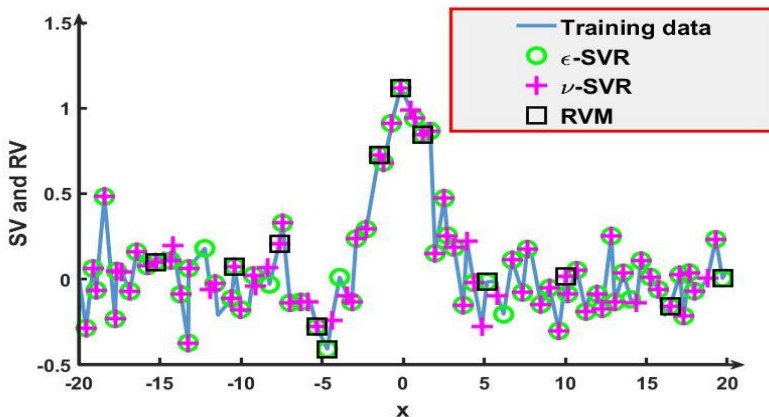

**Figure 6.** The number of RVs and SVs in Experiment (2).

**Table 4.** MSEs of prediction results for the three experiments.

| Algorithms | Experiment (1) | Experiment (2) | Experiment (3) |
|---|---|---|---|
| RVM | 0.0338 | 0.0305 | 0.0305 |
| $\varepsilon$-SVR | 0.0368 | 0.0431 | 0.0846 |
| $v$-SVR | 0.0348 | 0.0417 | 0.1000 |
| Percentage error for $\varepsilon$-SVR | 8.88% | 41.31% | 177.38% * |
| Percentage error for $v$-SVR | 2.96% | 36.72% | 227.87% * |

* Due to the tuning of three hyperparameters, the settings of the hyperparameters have certain challenges. The prediction results are overfitting.

**Remark 2.** *In this section, the research utilizes the prediction results of the Sinc function dataset to verify the performance of RVM. In the setting of the hyperparameters, the research designs different experiments according to the role of hyperparameters. In the process of experiments, the settings of C, $\varepsilon$, and $v$ have a great effect on the prediction accuracy and time consumed. After comparison, the prediction model obtained by RVM is relatively sparse (the number of training samples is 101). In Section 4.2, this research increases the number of training samples to further verify the sparsity and prediction performance.*

### 4.2. Prediction Results of Tank Test Dataset for KCS

In Section 4.1, using the Sinc function dataset, the sparsity and prediction performance of RVM are verified by the three experiments. The KCS is taken as the main research plant to further verify the sparsity and generalization of RVM. By dividing the training dataset and selecting the hyperparameters reasonably, the 3 DOF motion prediction models of KCS are obtained. The tank test data for the KCS is from the SIMMAN 2020 workshop [37]. It has some noise interference, so there is no need to add the noise interference. Table 5 shows the main particulars of KCS.

**Table 5.** Main particulars of KCS.

| Main Particulars | Values |
| --- | --- |
| Length between perpendiculars | 6.0702 m |
| Breadth | 0.8498 m |
| Mean draft | 0.2850 m |
| Block coefficient | 0.6505 |
| Displacement volume | 0.9565 m$^3$ |

In the SIMMAN 2020 workshop, the tank tests of KCS include the $-10°/-10°$, $20°/20°$, $-20°/-20°$ zigzag maneuvers, and the $-35°$ turning maneuver. Due to the effect of sensors and other factors, the test datasets have noise interference [37]. In addition, the training dataset should reflect more dynamic information of the KCS. Therefore, a small part of the $-10°/-10°$ and $-20°/-20°$ zigzag maneuver data is used to construct the training dataset. The $20°/20°$ zigzag maneuver data and $-35°$ turning maneuver data are taken as test datasets. The number of training and test datasets is shown in Table 6.

**Table 6.** Training and test datasets.

| Data Samples | Maneuvers | Number of Data Samples |
| --- | --- | --- |
| Training dataset | $-10°/-10°$ zigzag maneuver | 183 |
| | $-20°/-20°$ zigzag maneuver | 282 |
| Test dataset | $20°/20°$ zigzag maneuver | 618 |
| | $-35°$ turning maneuver | 700 |

As can be seen from Table 6, the number of training datasets is 465. This is about five times the number of training datasets in the Sinc function dataset. This research further verifies the sparsity of RVM from the perspective of KCS motion prediction. It is essential to determine the hyperparameters of the three algorithms to make the motion prediction results more accurate. These hyperparameters are set according to previous research [20,21,39]. For the prediction of $u$, $v$, and $r$, the hyperparameters of $\varepsilon$-SVR, $v$-SVR, and RVM are set to different values. The setting values of the hyperparameters are shown in Table 7.

**Table 7.** Setting of the hyperparameters for $u$, $v$ and $r$.

| Algorithms | $u$ | $v$ | $r$ |
| --- | --- | --- | --- |
| RVM | $\gamma = 0.09$ | $\gamma = 0.08$ | $\gamma = 0.01$ |
| $\varepsilon$-SVR | $C = 10^3$ $\gamma = 10^{-2}$ $\varepsilon = 10^{-3}$ | $C = 4 \times 10^3$ $\gamma = 10^{-2}$ $\varepsilon = 10^{-2}$ | $C = 150$ $\gamma = 10^{-2}$ $\varepsilon = 10^{-2}$ |
| $v$-SVR | $C = 10^3$ $\gamma = 10^{-2}$ $v = 0.5$ | $C = 10^3$ $\gamma = 10^{-4}$ $v = 0.5$ | $C = 400$ $\gamma = 10^{-3}$ $v = 0.5$ |

**Remark 3.** *Compared to ε-SVR and v-SVR, RVM only has one parameter to tune, and no optimization algorithm is needed for further optimization. This also reflects the advantages of RVM from the perspective of hyperparameter tuning. For SVR, the settings of C, ε, and v have a great influence on the results of ship motion prediction. Sometimes, the larger the C, the less accurate the prediction will be.*

According to the training datasets and the setting of hyperparameters, the motion prediction models of KCS are obtained. The test datasets are $-35°$ turning maneuver data and $20°/20°$ zigzag maneuver data, respectively. The prediction models trained by the three algorithms are compared to verify the sparsity. For the prediction models of $u$, $v$, and $r$, the number of RVs and SVs is shown in Table 8. Figure 7 shows the prediction results of $-35°$ turning maneuver data. Table 9 shows the MSEs of motion prediction results for the $-35°$ turning maneuver.

**Table 8.** The number of RVs and SVs for the tank test dataset.

| Algorithms | $u$ | $v$ | $r$ |
|---|---|---|---|
| RVM | 3 | 5 | 4 |
| ε-SVR | 264 | 37 | 93 |
| v-SVR | 251 | 239 | 239 |

**Table 9.** MSEs of prediction results for $-35°$ turning maneuver.

| Algorithms | $u$ | $v$ | $r$ |
|---|---|---|---|
| RVM | $6.40 \times 10^{-4}$ | $1.53 \times 10^{-4}$ | 0.2373 |
| ε-SVR | $2.72 \times 10^{-3}$ | $2.48 \times 10^{-4}$ | 0.3076 |
| v-SVR | $3.54 \times 10^{-3}$ | $1.87 \times 10^{-4}$ | 0.3702 |

RVM constructs a learning machine based on Bayes' theorem rather than the structural risk minimization principle. As the number of training samples increases, it can be seen from Table 8 that the number of RVs is far less than the number of SVs. This further verifies the sparsity of the proposed algorithm. For the trained model obtained by $v$-SVR, $v$ is set as 0.5 [21]. However, the sparsity of $v$-SVR is worse than that of ε-SVR. In Figure 7, the length of the absolute residual represents the prediction error. Moreover, only the absolute residuals of the ship motion prediction are presented. Figure 7a,b show that the prediction error of the three algorithms is small. As can be seen from Figure 7c, the three algorithms all have some prediction errors. For the prediction results of RVM, the maximum absolute residual is 0.52°. In the later stage of yaw angle prediction, the three algorithms all have certain deviations from Figure 7d. In the prediction results of the KCS motion trajectory, the degree of coincidence between the results predicted by the RVM and the tank test data is higher. For the time required to predict $u$, $v$, and $r$, RVM, ε-SVR, and $v$-SVR took the total of 0.749 s, 4.307 s, and 3.985 s, respectively.

Table 9 shows the MSEs of the prediction results. For the prediction results of $r$, the percentage errors for ε-SVR and $v$-SVR are 29.62% and 56.01%, respectively. Using the proposed algorithm, the accuracy of other prediction results can be improved by more than 22.22%. The generalization of RVM is demonstrated from the data prediction results. The $20°/20°$ zigzag maneuver data of the KCS is taken as another data sample to further validate the generalization of RVM. Although there are some similarities between the $20°/20°$ zigzag maneuver data and the $-20°/-20°$ zigzag maneuver data, there are some differences, such as the opposite excitation of the input rudder angle. The hyperparameter settings of the three algorithms are shown in Table 7. The prediction results of the $20°/20°$ zigzag maneuver are shown in Figure 8. Table 10 shows the MSEs of the prediction results for the $20°/20°$ zigzag maneuver.

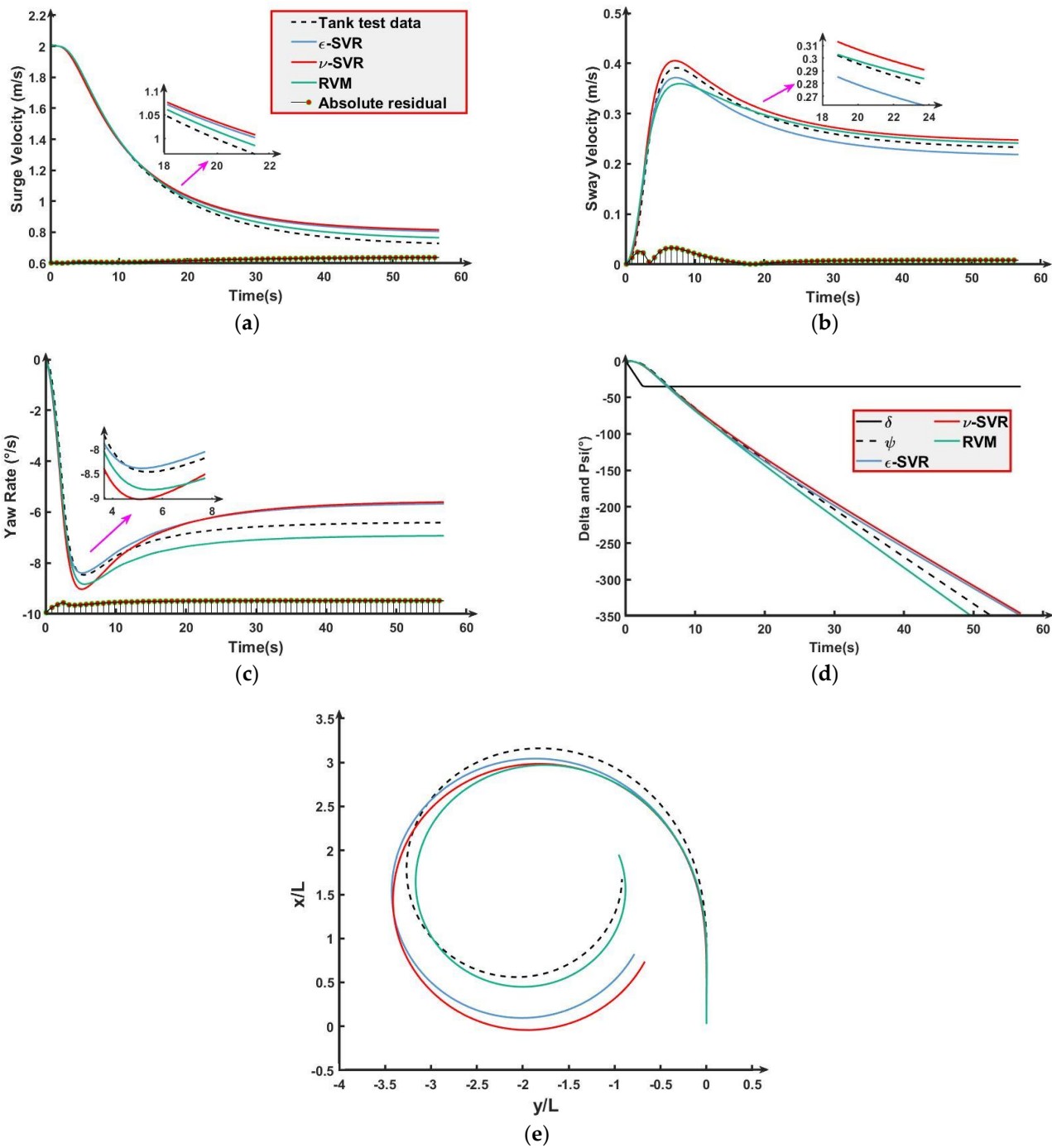

**Figure 7.** Prediction results of −35° turning maneuver: (**a**) is the prediction of surge velocity; (**b**) is the prediction of sway velocity; (**c**) is the prediction of yaw rate; (**d**) is the prediction of yaw angle; (**e**) is the prediction of ship motion trajectory.

It can be seen from Figure 8 that the three algorithms have high accuracy for motion prediction of the 20°/20° zigzag maneuver due to certain similarities between the test data and the training data. In Figure 8c,e, the prediction results of RVM have certain deviations in the later stage. However, as can be seen from Table 10, the overall prediction results of the KCS motion based on RVM are better than SVR. For the final prediction result, the accuracy can be improved by more than 14.04% with the proposed algorithm. For the time required to predict $u$, $v$, and $r$, RVM, $\varepsilon$-SVR, and $v$-SVR took the total of 0.746 s, 4.152 s, and 3.952 s, respectively. In addition, the three algorithms have high accuracy for $\psi$. The first

overshoot angle of the KCS is 33.56°. The first overshoot angle predicted by the proposed algorithm is 32.28°.

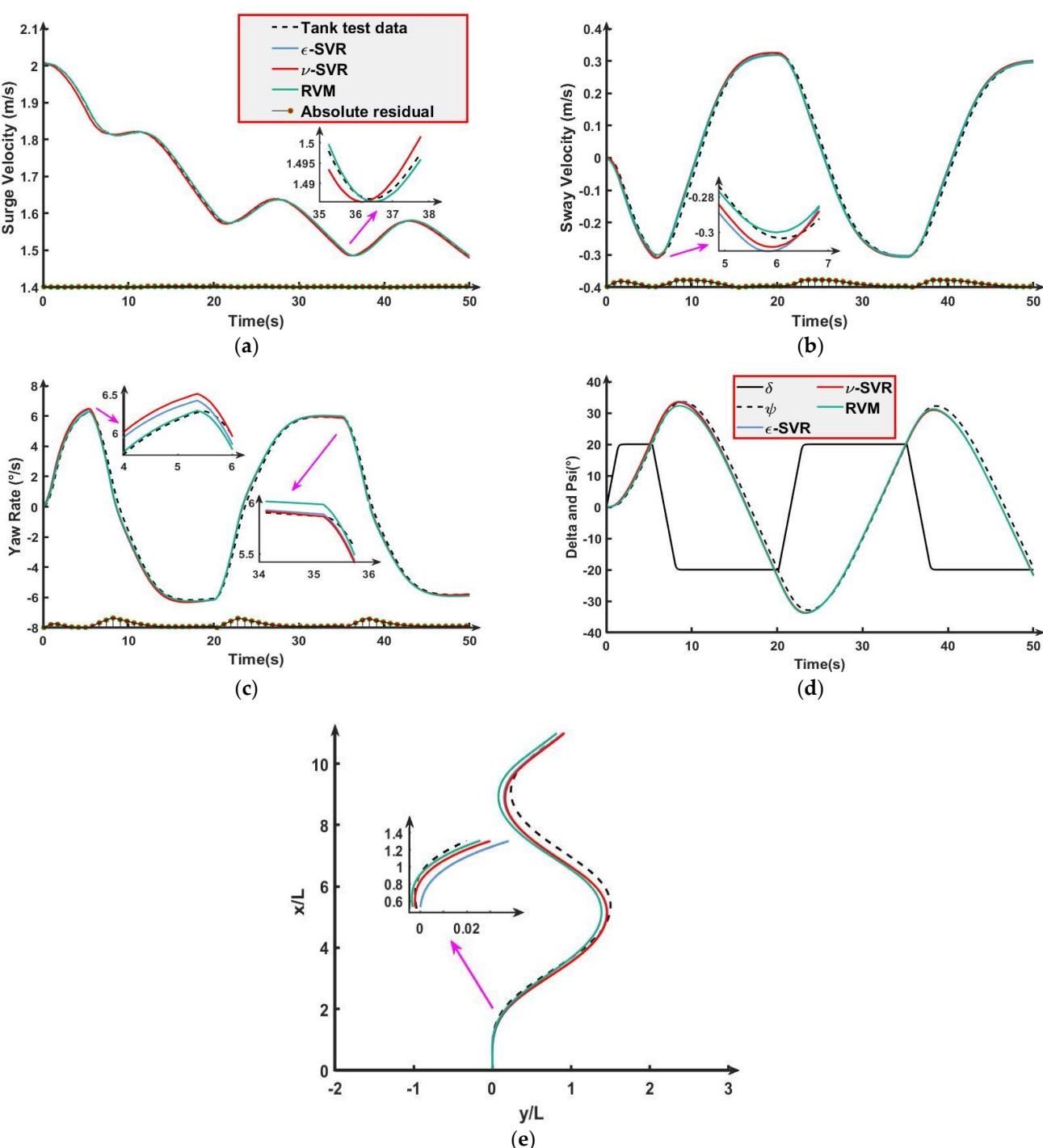

**Figure 8.** Prediction results of 20°/20° zigzag maneuver: (**a**) is the prediction of surge velocity; (**b**) is the prediction of sway velocity; (**c**) is the prediction of yaw rate; (**d**) is the prediction of yaw angle; (**e**) is the prediction of ship motion trajectory.

RVM is an algorithm that combines kernel methods and Bayes' theorem. Under the structure of a priori parameter, the automatic relevance determination is used to remove the irrelevant sample points to obtain a sparse model. The sparse model simplifies the model and can avoid overfitting to some extent. The reason is that it retains the explanatory variable most relevant to the response variable. In the prediction of −35° turning maneuver

data and $20°/20°$ zigzag maneuver data, the prediction models obtained by RVM have good sparsity and prediction accuracy.

**Table 10.** MSEs of prediction results for $20°/20°$ zigzag maneuver.

| Algorithms | $u$ | $v$ | $r$ |
|---|---|---|---|
| RVM | $3.27 \times 10^{-6}$ | $1.78 \times 10^{-4}$ | 0.0709 |
| $\varepsilon$-SVR | $3.55 \times 10^{-5}$ | $2.03 \times 10^{-4}$ | 0.0814 |
| $\upsilon$-SVR | $3.70 \times 10^{-5}$ | $2.06 \times 10^{-4}$ | 0.0813 |

**Remark 4.** *In this section, the research utilizes the prediction results of the tank test dataset to further verify the sparsity, effectiveness, and generalization of RVM. From the perspective of the trained model, as the number of training samples increases, the RVM can still maintain a certain degree of sparsity. Although there are certain prediction errors, the overall prediction curve of RVM fits the tank test data curve well.*

## 5. Conclusions

The main finding of this research is that RVM based on sparse Bayes' theorem is proposed to obtain a sparse and efficient ship motion prediction model. The final prediction results indicate that the proposed RVM can ensure good sparsity while ensuring prediction accuracy. Good sparsity of the trained model can reduce the prediction time, which is essential for the online prediction of ship motion. Therefore, the proposed scheme can provide a reference for online ship motion prediction. The experimental verification is performed based on the Sinc function dataset and tank test dataset of KCS, and the results are as follows:

(1) The Sinc function dataset is used as a simulation example to verify the sparsity and prediction accuracy of RVM. Based on the different hyperparameter settings of RVM, $\varepsilon$-SVR, and $\upsilon$-SVR, three experiments are conducted. The MSE of prediction results is lower than 0.0305 for RVM. The highest percentage of RVs in the training sample is below 17%. The prediction results of three experiments verify the sparsity, effectiveness, and generalization of RVM.

(2) Compared with the training dataset for the Sinc function, there are more training samples. However, although the number of training datasets has increased, the proposed scheme can still maintain sparsity. After dividing the training dataset and selecting the hyperparameters reasonably, the maneuvering motion prediction of the KCS is conducted. The accuracy of prediction results can be improved by more than 14.04%. The prediction performance of RVM is further verified.

(3) The prediction results of RVM are compared with those of $\varepsilon$-SVR and $\upsilon$-SVR. From the prediction results of two illustrative examples, all three algorithms have good prediction results, but the prediction performance of RVM is better than that of SVR. The overall prediction curve of RVM fits the tank test data curve well. Moreover, less time is also required. In addition, compared with the prediction models obtained by SVR, the prediction models obtained by RVM are sparser.

RVM has only one hyperparameter to adjust. Compared with SVR, the structure is simple. In terms of kernel function selection, RVM is not satisfied by the Mercer condition, so it has a wider selection of kernel function. It can also achieve a sparse prediction model and has the ability to predict quickly. However, for some more complex datasets, the prediction performance of RVM with a single kernel function remains to be studied. Ship motion prediction is of great significance for the design of course-keeping, path-following and adaptive controllers. In future, we will further improve the performance of the proposed algorithm and focus on ships' autonomous berthing and unberthing and path-following controllers using the improved algorithm.

**Author Contributions:** Conceptualization, Y.M. and X.Z. (Xianku Zhang); Methodology, Y.M. and X.Z. (Xianku Zhang); Validation, Y.M., X.Z. (Xianku Zhang) and X.Z. (Xiufeng Zhang); Writing—original draft preparation, Y.M.; Software, Y.M.; Writing—review and editing, X.Z. (Xianku Zhang), G.Z., X.Z. (Xiufeng Zhang) and Y.D.; Formal analysis, X.Z. (Xianku Zhang) and X.Z. (Xiufeng Zhang); Funding acquisition, X.Z. (Xianku Zhang); Supervision, X.Z. (Xianku Zhang), G.Z., X.Z. (Xiufeng Zhang) and Y.D. All authors have read and agreed to the published version of the manuscript.

**Funding:** This work is partially supported by the National Key R&D Program of China (Grant No. 2022YFB4301402), National Science Foundation of China (Grant No. 51679024), Dalian Innovation Team Support Plan in the Key Research Field (Grant No. 2020RT08), Fundamental Research Funds for the Central Universities (Grant No. 3132021139, 3132023137, 3132023502) and Doctoral Research Initial Fund Project of Liaoning Province (No. 2021-BS-078).

**Institutional Review Board Statement:** Not applicable.

**Informed Consent Statement:** Not applicable.

**Data Availability Statement:** The data used to support the findings of this study are available from the corresponding author upon request.

**Acknowledgments:** Much appreciation is given to each reviewer for their valuable comments and suggestions to improve the quality of this writing. The authors would like to thank anonymous reviewers for their valuable comments to improve the quality of this article.

**Conflicts of Interest:** The authors declare no conflict of interest.

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
