# Peer review of "Sparse Bayesian Relevance Vector Machine Identification Modeling and Its Application to Ship Maneuvering Motion Prediction"

_jmse, doi:10.3390/jmse11081572_

Round 1

Reviewer 1 Report

The paper is devoted to the problem of Spars Bayesian relevance vector machine identification modeling with its application to ship maneuvering motion prediction.

The paper is of interest.

The introduction of the paper is written in detail and reveals the direction of further research.

The mathematical model of the investigated object - KRISO container ship is given.

However, Equation (1), describing the motion model with 3 degrees of freedom, contains both exclusion parameters (for example, m) and time-varying parameters (for example, X, Y, N). For all time-dependent parameters, the notation X(t), Y(t), N(t) is recommended to use.

The above methodology allows us to understand the idea of solving the problem.

Illustrative examples provided.

That being said, the paper makes it unclear how the Sinc function dataset is used. Is the set of 201 points the ship's assigned course, or is it a course deviation? What justifies the choice of precisely 201 dataset points?

Subsequently, the authors divide the Sinc function dataset into test and test sets in the 50% / 50% ratio. However, at least 70% of the data is often used for the training set and the remaining 30% or less for the test set. Are 101 points of data enough to build a model?

In the paper (if it’s possible), I would like to see information on how the simulation was carried out, using what software.

Author Response

Thanks for your valuable suggestion and righteous reviewing of this manuscript. We have endeavored to make the revision of this manuscript under the supervision of the reviewers. In the Response Letter, we have given some reasonable explanations to the professional comments put forward by the reviewers, hoping to be recognized by the professional reviewers. The attachment is the detailed response to your comments. Thanks again for your work.

Reviewer 2 Report

In order to establish a sparse and accurate ship motion prediction model, a Bayesian probability prediction model based on relevance vector machine(RVM) is proposed for the nonparametric modeling.

there are many papers in section 1.2. however, contributions of your paper compared to the state of the art on ship motion prediction model are not clear to me.

In section 1.1, you mention that "The sparse modeling method is able to overcome overfitting to a certain extent and reduce the prediction time, which can lay the foundation for online prediction. " can you present the reason or references for this arguement? 

You mention that online (real-time) prediction of ship motion is important. can you state why online prediction is important? what is the usage of the online prediction of ship motion?

can you predict the ship motion in real time? what is the computational load of your method?

SVR needs to determine hyperparameters, such as penalty factor(?). Their setting has certain effect on the prediction performance of SVR. In  order to overcome the effect, Zhang et al. [19] and Zhu et al. [30] used swarm intelligence algorithm to tune the hyperparameters to get good prediction results. Can you compare your method with [19,30]? also, computational load needs to be compared with [19,30]

ok

Author Response

(The authors gave the same response as above.)

Reviewer 3 Report

The authors presented innovative research in the field of ship maneuvering motion prediction. However, the manuscript as it stands does not constitute a scholarly article useful to readers.

Comments:

1. The Introduction lacks the formulation of the thesis of the work and the resulting goals, especially with an indication of its practical usefulness.

2. In line 126, the source of equations (1) and the justification for their selection from among many possible in the literature of ship hydromechanics or control engineering are not given.

3. Description of the components of equation (2) should be supplemented with the sources of forces and moments coming from controls and disturbances.

4. Focus the content of the entire manuscript not on your original research achievement, but on the reader, how much will benefit from it.

5. Conclusions lacks a detailed plan for further research, in particular practical applications in shipbuilding and MASS automation.

Author Response

(The authors gave the same response as above.)

Round 2

Reviewer 2 Report

I am satisfied with revision

Ok

Reviewer 3 Report

Since the authors of the manuscript took into account all my comments to improve its substantive quality from the reader's point of view, I propose to accept it for publication in the Journal of Marine Science and Engineering.